# Avoiding the bullies: The resilience of cooperation among unequals

**Michael Foley**[1], **Rory Smead**[2], **Patrick Forber** [3], **Christoph Riedl** [1,4,5,6]*

**1** Network Science Institute, Northeastern University, Boston, Massachusetts, United States of America, **2** Department of Philosophy and Religion, Northeastern University, Boston, Massachusetts, United States of America, **3** Department of Philosophy, Tufts University, Medford, Massachusetts, United States of America, **4** D'Amore-McKim School of Business, Northeastern University, Boston, Massachusetts, United States of America, **5** Khoury College of Computer Sciences, Northeastern University, Boston, Massachusetts, United States of America, **6** Institute for Quantitative Social Science, Harvard University, Cambridge, Massachusetts, United States of America

* c.riedl@neu.edu

**Data Availability Statement:** Replication code available on GitHub: https://doi.org/10.5281/ zenodo.4583746.

**Funding:** This research was supported in part by Office of Naval Research grants N00014-16-1-

## Abstract

Can egalitarian norms or conventions survive the presence of dominant individuals who are ensured of victory in conflicts? We investigate the interaction of power asymmetry and partner choice in games of conflict over a contested resource. Previous models of cooperation do not include both power inequality and partner choice. Furthermore, models that do include power inequalities assume a static game where a bully's advantage does not change. They have therefore not attempted to model complex and realistic properties of social interaction. Here, we introduce three models to study the emergence and resilience of cooperation among unequals when interaction is random, when individuals can choose their partners, and where power asymmetries dynamically depend on accumulated payoffs. We find that the ability to avoid bullies with higher competitive ability afforded by partner choice mostly restores cooperative conventions and that the competitive hierarchy never forms. Partner choice counteracts the hyper dominance of bullies who are isolated in the network and eliminates the need for others to coordinate in a coalition. When competitive ability dynamically depends on cumulative payoffs, complex cycles of coupled network-strategy-rank changes emerge. Effective collaborators gain popularity (and thus power), adopt aggressive behavior, get isolated, and ultimately lose power. Neither the network nor behavior converge to a stable equilibrium. Despite the instability of power dynamics, the cooperative convention in the population remains stable overall and long-term inequality is completely eliminated. The interaction between partner choice and dynamic power asymmetry is crucial for these results: without partner choice, bullies cannot be isolated, and without dynamic power asymmetry, bullies do not lose their power even when isolated. We analytically identify a single critical point that marks a phase transition in all three iterations of our models. This critical point is where the first individual breaks from the convention and cycles start to emerge.

3005 (CR) and N00014-17-1-2542 (CR). The funders had no role in study design, data collection and analysis, decision to publish, or preparation of the manuscript.

**Competing interests:** The authors have declared that no competing interests exist.

## Author summary

Individuals often differ in their ability to resolve conflicts in their favor, and this can lead to the emergence of hierarchies and dominant alphas. Such social structures present a serious risk of destabilizing cooperative social interactions or norms. Why work together to find food when a more aggressive or stronger individual can take all of it? In this paper we use game theory and agent-based modeling to investigate how cooperative behavior evolves in the presence of powerful bullies who have no incentive to cooperate. We show that when individuals can choose their interaction partners, bullies do not always destabilize cooperation. Instead, cooperative norms survive as individuals learn to avoid dominant individuals who become isolated in the population. When competitive ability itself depends dynamically on past success, complex cycles of coupled network-strategy-rank changes emerge: effective collaborators gain popularity and thus power, adopt aggressive behavior, get isolated, then lose power. Our results have important implications: in our modeled scenario the rich do not always get richer, the dominance of bullies can be broken, and inequality in accrued resources can be eliminated. Thus, our work provides new insight into potential sources of, and strategies for avoiding, resource inequality.

## Introduction

Individuals often differ in their ability to fight and win conflicts over contested resources. Power asymmetries in conflicts can lead to the emergence of hierarchies and dominant alphas. Such social structures present a serious risk of destabilizing cooperative social interactions or norms. Dominant individuals—bullies, in effect—undermine collaboration among other individuals and are very dangerous to confront. Why work together to find food when the stronger individual can simply take all of it?

Yet our human ancestors were nomadic, lived in groups (individuals did not hold territories), and foraged via collaborative interactions [1, 2]. Human cooperation and collaboration has only accelerated since, especially through trade and information sharing. When resources are gained through social interactions, cooperation is threatened by bullies who have no incentive to cooperate. The presumed evolutionary solutions to this problem involve coalition-building where individuals organize to collectively punish, exile, or kill the destabilizing individual [3–5]. Coalition-building can be challenging because it requires coordination and communication—tools that require the stability of cooperation to emerge. We seek to investigate whether and how bullies can be controlled or managed in this scenario without presuming rich coordination and communication skills. We develop a model using evolutionary game theory where payoffs are generated during social interaction rather than by holding territories or individual foraging, and individuals can choose their interaction partner. Our model identifies a pathway for dealing with powerful individuals that does not require collective action: partner choice allows individuals to isolate bullies and mitigate their damaging effects without the need for explicit coordination, communication or agreement. Furthermore, when power asymmetry dynamically depends on cumulative payoffs, complex interdependent cycles emerge among strategies, network connections, and ranks.

We use a generalized hawk-dove game to represent the strategic interaction involved in a competition over a contested resource [6–8]. In this game of conflict, players simultaneously choose either an aggressive *hawk* or deferential *dove* strategy. Playing *hawk* against an opponent playing *dove* secures the resource and yields the best payoff. However, playing *hawk* against another *hawk* results in a costly conflict and the worst payoff. The *dove* strategy is the

safer option against *hawk* opponents: avoiding fights with the *hawks* by relinquishing the resources and sharing the resource with other *doves* (see S1 Text for technical details).

The usual (evolutionarily stable) solution to the generalized hawk-dove game can be interpreted as an equilibrium in which individuals randomize between *hawk* and *dove* strategies. The randomizing solution results in a significant number of costly *hawk-hawk* conflicts. However, if individuals can use an external cue (such as whether they are the territory owner or intruder) to coordinate their actions, then more efficient new solutions emerge. These correlated equilibria avoid conflict entirely, outperform the randomizing strategy, are stable, and can easily invade a population [7, 9–12]. Since these correlated equilibria identify patterns of behavior that are "customary, expected, and self-enforcing" [13], they are identified as paradigm examples of conventions in game theory [14, 15]. Two equally effective conventions are possible: individuals may behave aggressively when they are hosts and defer when they are visitors (a sort of ownership norm) or vice versa (a host-guest hospitality norm). Whichever convention emerges first takes over. Many studies have focused on the ownership norm (the so-called "bourgeois" solution) since it appears more prevalent in nature [7, 10, 14, 16]. However, partner choice has the striking effect of favoring a host-guest norm (also called the "paradoxical" convention) instead of the ownership norm [12]. Although there is rich work in this area, there needs to be a formal investigation into the interaction between power asymmetry and partner choice, and instances when power asymmetry can dynamically depend on cumulative payoffs.

Partner choice is an important and realistic mechanism. Social networks change continuously as individuals form new ties and dissolve existing ones [17–19]. Individuals modify their social surroundings by choosing who to interact with and how much time to allocate to each interaction [20, 21]. It is now increasingly clear that the heterogeneous structures we observe in empirical social networks are the result of an interplay between behavior and partner choice [22, 23]. Partner choice gives rise to interaction networks that change over time. We adopt the term dynamic network for our partner choice model; others have used the term "temporal" or "evolving networks" [24].

Building on previous dynamics networks studies [12, 25, 26] and two recent studies that investigate inequality in humans using public goods games [27, 28], we introduce power asymmetries into games of conflict played on dynamic networks. Power takes the form of differences in competitive ability and has the implication that different individuals may effectively be playing different games. Formulations of similar asymmetric games in which individuals receive different payoffs or face different games have been studied in the context of static networks [29] and two-player interactions [30].

Using the hawk-dove framework, we model power asymmetries by assigning individuals ranks and supposing that ranking individuals secure an additional positive reward ($f$) in instances of *hawk-hawk* conflicts. Without loss of generality, if $i$ outranks $j$ the payoffs are given by

$$
\begin{array}{cc}
& \text{hawk}_j \quad \text{dove}_j \\
\begin{array}{c} \text{hawk}_i \\ \text{dove}_i \end{array} &
\left( \begin{array}{cc}
f, hh & hd, dh \\
dh, hd & dd, dd
\end{array} \right)
\end{array} \tag{1}
$$

with $hd > dd > dh > hh$ and $f \geq hh$.

The value of this additional reward ($f$) for ranking individuals in conflicts is a central variable in our study. It significantly affects whether the correlated conventions are resilient in the presence of bullies. Partner choice also has an important effect. We find that when individuals interact randomly, the convention persists even under substantial power asymmetry, but it

eventually disappears completely and the population breaks into pure *hawks* and *doves*. However, with partner choice, the convention remains stable under any level of power asymmetry —aggressive bullies are isolated and the convention is preserved in the population. Finally, if ranks themselves dynamically depend on accumulated payoffs, we see the emergence of cycles in which individuals move through the ranks, adapt their strategy, and change network position.

There is much literature on the evolution of cooperation in randomly mixing populations, structured populations, on graphs and networks, and on dynamic networks (see [31, 32] for reviews). Our work provides three advances over prior work. First, the focal game in the literature on dynamic networks has predominantly been the well-known Prisoner's Dilemma. Here we examine games of conflict which naturally lend themselves to the study of inequality and exploitation. Network structure has been shown to be more important in games of conflict than the Prisoner's Dilemma [33], which warrants additional analysis of this game in connection with partner choice. Second, prior work on games of conflict has mostly focused on randomly mixing populations or static networks, thus eliminating the critical mechanism of partner choice. There are notable exceptions that study games of conflict on dynamic networks [12, 34, 35], but in these models either individuals cannot use an external cue to coordinate their actions (i.e., the models do not allow for correlated equilibria as solutions) or they do not investigate power asymmetry. Previous models also do not include both asymmetric power and partner choice [36, 37]. Third, models that do include power inequalities [6, 8, 10, 16] do not consider networked interaction or partner choice, and they assume a static game where a bully's advantage does not change over time. We summarize the most closely related prior work in Table A in S1 Text.

## Results

We explore our model (see Methods) analytically and using agent-based simulations. Each round, individuals independently choose an opponent to play in a game of conflict (Fig 1). They choose a strategy to play in the interaction and receive a payoff (Eq 1). They can distinguish whether they initiate the interaction (i.e., are the visitor) or not (i.e., are the host). Learning occurs in our agent-based simulations through simple yet psychologically realistic Roth-Erev reinforcement learning [26, 38, 39]. Agents update both who they visit as well as their strategies, distinguishing between hosting and visiting according to the accumulated payoffs of each strategy in their respective role. As a result, the structure of the social network coevolves along with the behavior of individuals. Our key modeling parameter is the degree of power asymmetry *f*, which we vary between 0 (no power asymmetry, ranks play no role) and 1 (large asymmetry, ranking individual can always receive the maximum payoff). All main results are robust with respect to population size and variation in learning speed (see S1 Text).

We consider three iterations of our model. First, power asymmetries in the random interaction case (i.e., static, fully connected networks with uniform tie weights). Second, power asymmetries with partner choice (i.e., dynamic networks). Third, dynamic power asymmetries that evolve as a function of cumulative payoffs with partner choice.

### Power asymmetries and random interaction

When individuals interact randomly, the introduction of power asymmetry has a dramatic impact: we see the emergence of top-ranked individuals who no longer follow the convention —bullies—and the eventual collapse of the correlated convention for the entire population. Whether the correlated convention breaks (and for which individuals) depends critically on the value of *f*, the payoff to the ranking individual in *hawk-hawk* interactions. To understand

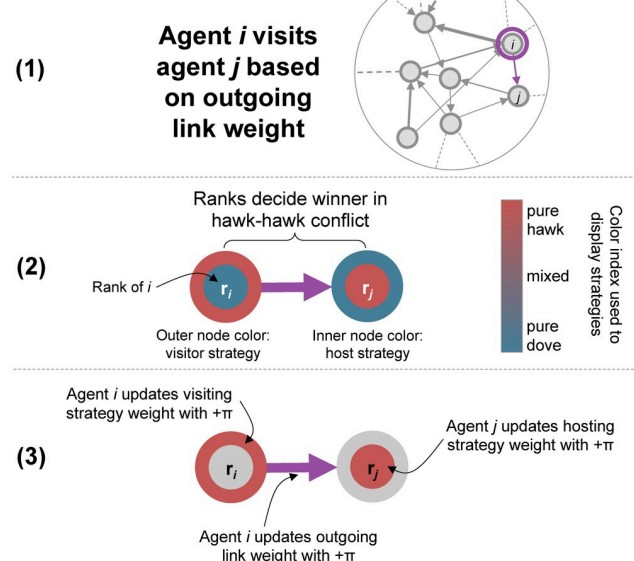

**Fig 1. Game play and updating mechanism.** In each round agents must (1) select an interaction partner, (2) play a game of conflict, and (3) update both their network weight and strategy weights based on the payoff $\pi$.

why and how this happens we examine power asymmetries with random interaction both analytically (Fig 2A and S1 Text) and numerically through agent-based modeling (Fig 2B).

A simple analytic model provides some insight into how power asymmetry affects correlated conventions. Below the critical value of $f < dh$, agents of all ranks face, on average, a

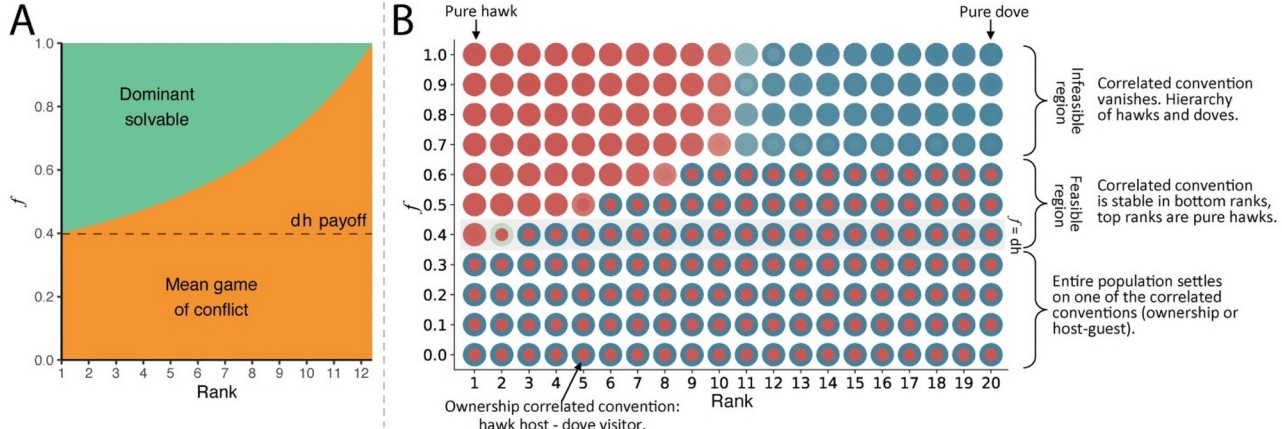

**Fig 2. In games of conflict with power asymmetry and random interaction, individuals with different ranks effectively face different games. A**, Analytical analysis of our model reveals that, under random interaction, individuals play different games, depending on their rank and the degree of the power asymmetry $f$ (see S1 Text). For $f < dh$, power asymmetry has no effect and individuals of all ranks engage in a mean game of conflict (the orange area). For $f \geq dh$, the game is dominant-solvable for the top ranked individual(s). **B**, Numerical results from evolutionary simulations show that conventions prevail when $f$ is below the critical value $dh$; individuals of all ranks play correlated conventions (either all ownership or host-guest; each appear with equal frequency in simulation seeds; ownership shown). At $f \geq dh$ a transition occurs as ranking individuals break from the convention and adopt pure *hawk* behavior both home and away. The correlated convention is resilient among most individuals until high values of $f$ make it unsustainable. A hierarchy forms in which the top-half of individuals are pure *hawks* while the bottom-half are pure *doves*. Note how the analytic result on the left matches the numeric result on the right. (Both panels use $n = 20$; $dh = 0.4$; $dd = 0.6$).

game of conflict ($f = 0$ corresponds to the baseline symmetric game of conflict). At the critical value of $f \geq dh$, *hawk* becomes a dominant strategy for at least one of the ranking players, while the outranked player's best strategy depends on the chance an opponent plays *hawk* (see S1 Text) for the full derivation). Critically, this transition does not depend on population size or any other parameters of the model besides the *dh* payoff.

As *f* increases beyond the threshold, the number of ranking individuals for whom the game is dominance-solvable increases. The convention becomes unsustainable at high values of *f*. If the expected probability that an opponent will play *hawk* ($P(hawk_j)$) rises above a certain value, then bottom-ranked individuals will prefer to play *dove* as both visitors and hosts. That value depends on the payoffs of the game and the chance of outranking an opponent (given by *R*; see S1 Text for the full derivation):

$$P(hawk_j) > \frac{hd - dd}{(dh - dd) - (R \cdot f - hd)}. \tag{2}$$

The expectation for opponent $P(hawk_j)$ depends on the proportion of the population for whom the game is dominance-solvable. Moving from random interaction to our model with partner choice has a significant effect on this expectation and on *R*.

With the insight from the analytic model in hand, we turn to the main analysis of our work using agent-based simulations (Fig 2B). These simulations bear out the analytical results. For values below the critical value of $f < dh$, the entire population settles on one of the two correlated conventions (with equal proportions of simulations settling into either the ownership or host-guest equilibrium). At the critical value of $f \geq dh$, the transition occurs where the top-ranked individual stops adhering to the correlated convention. Bullies start to adopt the aggressive strategy as both hosts and visitors, while outranked individuals continue to settle on one of the correlated conventions. Interestingly, if $f > dh$ but does not significantly exceed it, the bottom-ranked individuals do not adopt pure-dove strategies even if outranked by everyone else. The reason is that most mid-ranked individuals learn to play *dove* when hosting in response to visits from ranking individuals. Once there are a significant number of individuals who play *dove* as hosts, this allows the bottom-ranked individuals to be successful playing *hawk* when they visit despite always being out-ranked by their hosts. This also explains why we see individuals adopt mostly pure strategies: since they can always coordinate on a correlated convention there is no need to randomize between strategies. As *f* increases, the number of bullies (individuals who always play *hawk* both at home and away) increases, reducing the number of individuals playing the correlated convention. Simulations show that the convention breaks for populations of size twenty near $f = 0.7$. A two-class system emerges where individuals ranked in the top half of the population settle on pure *hawk*, while those in the bottom half settle on pure *dove*.

## Power asymmetries and partner choice

Partner choice profoundly affects individual behavior in equilibrium. The correlated convention never breaks on dynamic networks—even for the highest degree of power asymmetry—and the cooperative convention remains stable for most of the population (Fig 3A). Those results are robust across different population sizes (Fig 3B). The key mechanism behind this result is that partner choice allows individuals to avoid interacting with bullies. The same phase transition as happened in the case of random interaction occurs at $f \geq dh$ when a few top ranked individual(s) adopt aggressive strategies while the correlated convention is preserved among the rest of the population. However, contrary to the random interaction case, the second transition never occurs. The correlated convention never breaks, even at very high

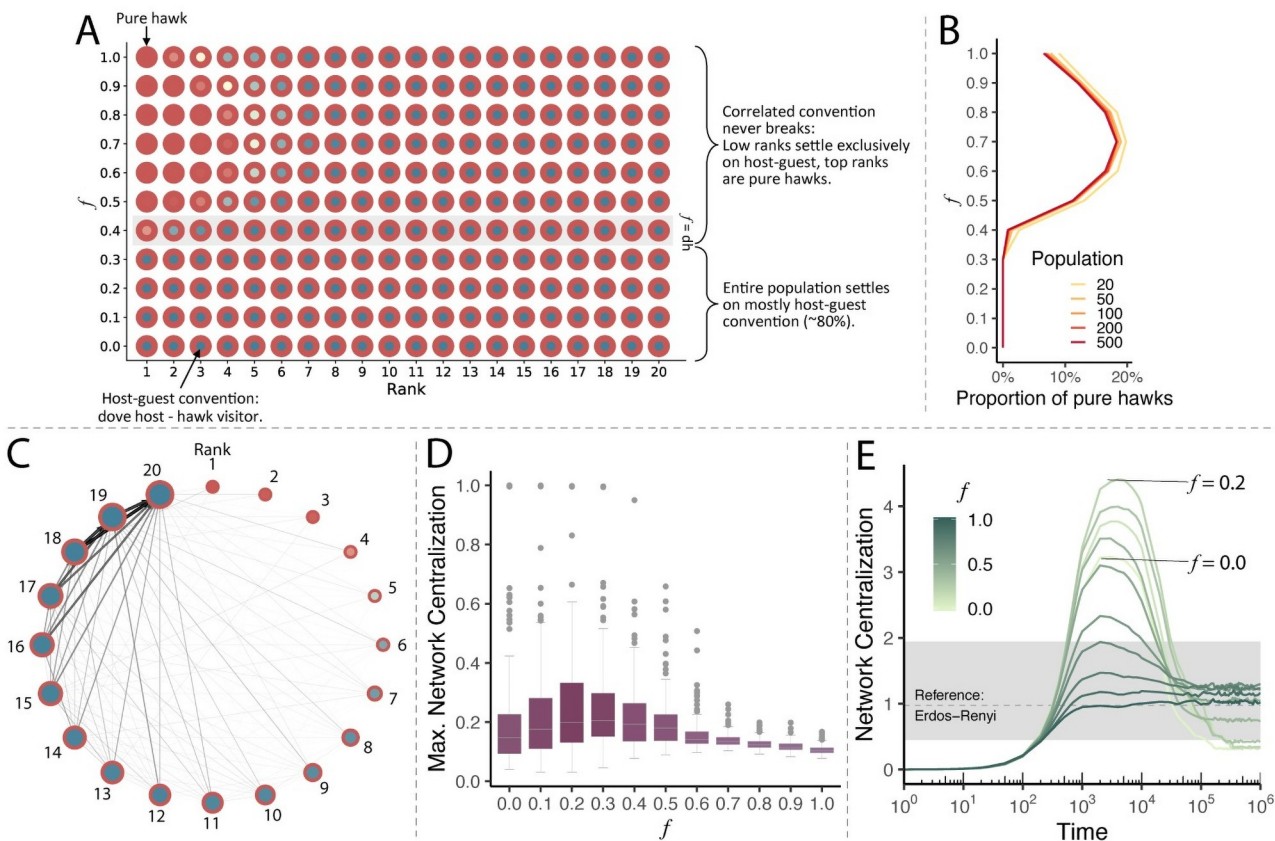

**Fig 3. Partner choice restores the correlated convention and increases cooperation in the presence of bullies. A**, With partner choice and low *f*-value, individuals of all ranks play correlated conventions (predominantly host-guest as first shown in [12]). A transition occurs at $f \geq dh$, identical to the random interaction case, where top ranked individual(s) break from the convention and become pure *hawks*. The correlated convention is preserved among outranked individuals. However, contrary to the random interaction case, partner choice allows the convention to remain sustainable among a majority of individuals even for high *f* values. This is infeasible under random interaction. **B**, The pattern in which some top ranked individual (s) break from the convention is consistent across population sizes. Notice how the shape of the curve in **B** matches the boundary where strategies change in **A**. Pure *hawks* are defined conservatively as agents with a likelihood of at least 0.8 for playing *hawk* both at home and away. **C**, The network structure that emerges resembles a hierarchy. Ranking individuals play pure *hawk* strategies but receive few or no visitors at all. Outranked individuals adopt the correlated convention and attract many visitors. The individual who is outranked by all others is visited by everyone. The graph shows average in-weights across seeds grouped by rank. Node size is scaled by incoming edge weight ($dh = 0.4$; $dd = 0.6$; $f = 0.6$). **D**, The highest degree of network centralization (most hub-like) is reached at $f = 0.2$. Nodes with disproportionate (too many) connections stop emerging entirely at $f = 0.8$ (color by mean network centralization). **E**, Network centralization over time, averaged across seeds. Networks with heterogeneous node weights emerge for a period of time. Network centralization of random Erdős–Rényi networks of the same size and density are shown as reference (the dashed line shows the median, the grey area is 95% CI).

values of *f*. Partner choice restores the stability of the convention because individuals learn to visit those they outrank and to play *hawk* when they do so. As a result, a competitive hierarchy forms (Fig 3C). The ranking individual adopts aggressive strategies both at home and away, does not care who to visit, but hosts no visitors in return. At lower ranks, individuals target an increasingly smaller set of others who they outrank. They receive visitors in proportion to the number of others who outrank them and tend to play dove at home because they tend to be visited by individuals playing *hawk*. Consequently, very few *hawk-hawk* conflicts occur despite the increased incentive—the increased *f* value—for ranking individuals. As in the random interaction case, this allows even the lowest-ranking individual to play *hawk* when visiting as there are always available *dove* hosts to visit. The presence of a few aggressive bullies preserves

the correlated convention for all other individuals: avoiding the bullies changes how the rest interact.

We find the highest proportion of pure *hawks* for intermediate values of $0.6 < f < 0.8$, while the number of pure *hawks* decreases for very high *f* values (Fig 3B). The reason for this is that as *f* approaches 1, the alpha (top-ranking) individual stops distinguishing among all other individuals. This individual simply learns to always play *hawk* and all possible hosts are seen as equivalent. At the same time, additional early hawk behavior from ranking individuals may lead lower-ranked individuals to avoid visiting. Consequently, ranking individuals (with the exception of the alpha) have higher proportions of *hawk* visitors compared to lower *f* and therefore have a greater incentive to play dove when hosting. Pure *hawks* reach a maximum of around 20% of the population at $f = 0.7$ with only around 15% of pure *hawks* at $f = 0.9$. This shows that excessive power asymmetry is not always detrimental. It can in fact increase cooperation in a population by helping preserve the correlated convention.

Introducing power asymmetry into a population affects which of the two possible correlated conventions emerges—the ownership or host-guest norm. The population *always* settles on the host-guest equilibrium rather than the ownership equilibrium after the first phase transition. The reason is that with partner choice, all but the lowest ranked individual can locate an even lower-ranked individual to act aggressively toward. This means all visitors will learn aggressive strategies and that the optimal response as a (typically) outranked host with aggressive visitors is to play dove. At the same time, the ranking individual adopts aggressive strategies and does not discriminate among which other individuals to visit. This ensures that everyone except the ranking individual receives at least some visits from a ranking individual who cannot be beaten in conflicts, thereby incentivizing individuals to play dove as hosts. The ownership solution would require that all outranked individuals play dove when visiting, but that behavior is not stable as even the second-lowest ranked individual can pursue the aggressive strategy against the lowest-ranked individual and secure the *f* payoff by playing aggressively. As a result, only the host-guest equilibrium is stable. Note that in the case of $f = 0$—the baseline case in which ranks do not matter—the correlated convention that emerges is also predominantly host-guest rather than ownership. This is the main finding reported in [12].

We analyze the structure of the emerging interaction networks. Remember that our networks are weighted, with tie weights updated through reinforcement learning. Our analysis of network structure hence focuses on the analysis of the distribution of in-weights (i.e., how many visitors an individual is expected to receive), rather than the discrete number of connections (node degree). We quantify network heterogeneity which we label *Network Centralization* as the variance of in-degree network weights $Var(\Sigma_j w_{j1}, \Sigma_j w_{j2}, \ldots, \Sigma_j w_{jn}) \forall j \in N \& j \neq i$ (see also S1 Text). High values of network centralization indicate the presence of "hubs": nodes with disproportionately strong in-weights compared with the average value. We show the distribution of the maximum network centralization at any point during the evolution (Fig 3D). The network structures that emerge critically depend on *f*. For low *f* values, we find large variation in the types of networks that emerge, ranging from mostly homogeneous networks to centralized hub-and-spoke networks (Fig B in S1 Text). When ranks play only a minor role, individuals have no strong preferences for who to visit and a variety of equilibrium solutions are possible as reported in [12]. In those cases, networks range from almost completely homogeneous to fully star-like. For high degrees of power asymmetry, individuals have clear preferences who to visit and a hierarchical social network forms that is similar across all simulations (Fig 3C). Thus, the highest network centrality of any individual is reached at $f = 0.2$ and it is generally lower for higher values of *f*. For $f \geq 0.7$, centralized structures are virtually absent. Strong hierarchies form with each individual visiting all lower-ranked individuals with equal probability.

We show network centralization averaged across seeds over time along with a random Erdős–Rényi networks of the same size and density as reference (Fig 3E). The variance of the network weights at initialization is 0 (all weights are uniformly set to $\frac{L}{N-1}$; $L = 19$). For comparison, the expected variance of a Erdős–Rényi random network of size $N = 20$ is 1.12, while the variance of a $N = 20$ star network is 17.95. Network centralization increases over time with a significant spike in centralization at around 5,000 time steps (Fig 3E and Fig B in S1 Text). That is when we see outranked individuals turning into highly connected hubs that attract many visitors. Hubs disappear once the convention is established: there is no reason to visit the *dove* hub anymore as most other individuals play *dove* as hosts as well. If $f < dh$, networks converge to a nearly completely homogenous state and maintain more heterogeneity when $f \geq dh$. For a period during the evolution (roughly between time 2,000 and 50,000) the networks that emerge for low-to-middle $f$ values share qualitative and quantitative properties with those that we observe in human social networks. There are a few nodes with disproportionate (too many) connections, while most nodes have few connections [40–42].

Is the convention restored even when partner choice and strategies evolve on different timescales? Previous studies show that the timescale of strategy and partner choice dynamics can greatly alter the cooperation level [39, 43, 44]. Here however, we find that outcomes do not depend crucially on the ratio between the partner choice and strategy updating timescales. Faster partner choice dynamics increase the proportion of host-guest interactions relative to the equal speed case and shorten the overall time needed for simulations to converge (Fig D in S1 Text). Slower partner choice dynamics reduce the number of host-guest interactions somewhat and lengthen the time needed to converge. However, even with very slow partner choice updating, the convention continues to emerge and the vast majority of interactions are host-guest (e.g., around 98% in the $f = 0.7$ case). The convention continues to survive even if network learning speed is at $\frac{1}{100000}^{th}$ of partner choice speed (Fig E in S1 Text). Thus, even very slow partner updating is sufficient to allow populations to isolate bullies and thus preserve the correlated convention.

## Dynamic power asymmetries

So far, we have restricted our analysis to instances in which power asymmetries—i.e., the ranks of individuals—are determined randomly and remain fixed over time. However, in the real world, power does not remain fixed and may depend on an individual's accumulated wealth or resources [45, 46]. We address this possibility by allowing agents' ranks to change dynamically as a function of total cumulative payoffs (see Methods). Individuals of different rank not only face different games (as shown in Fig 2A) but they do so dynamically as their rank changes. Allowing rankings to be dynamic results in striking cycles (Fig 4), but the correlated convention remains resilient.

Ranking individuals do well in aggressive interactions which they exploit—they adopt increasingly aggressive strategies and become bullies. Doing so leads others to learn to avoid them, which gives bullies less access to shared resources from collaboration. As a result, their rank drops and the success of the aggressive strategy fades. As their rank falls, they learn to adopt a more cooperative strategy. They become part of a cooperative cluster that has coordinated on the correlated convention. As these individuals become *dove*-host hubs, thereby attracting many visitors, they accumulating high total payoffs. They climb in the rankings as a result, and the cycle repeats. Rank, network position, and strategy change in parallel such that adopting an aggressive strategy is quickly followed by a sharp drop in visitors (becoming a network spoke), which leads to a steady decrease in rank (Fig 5A).

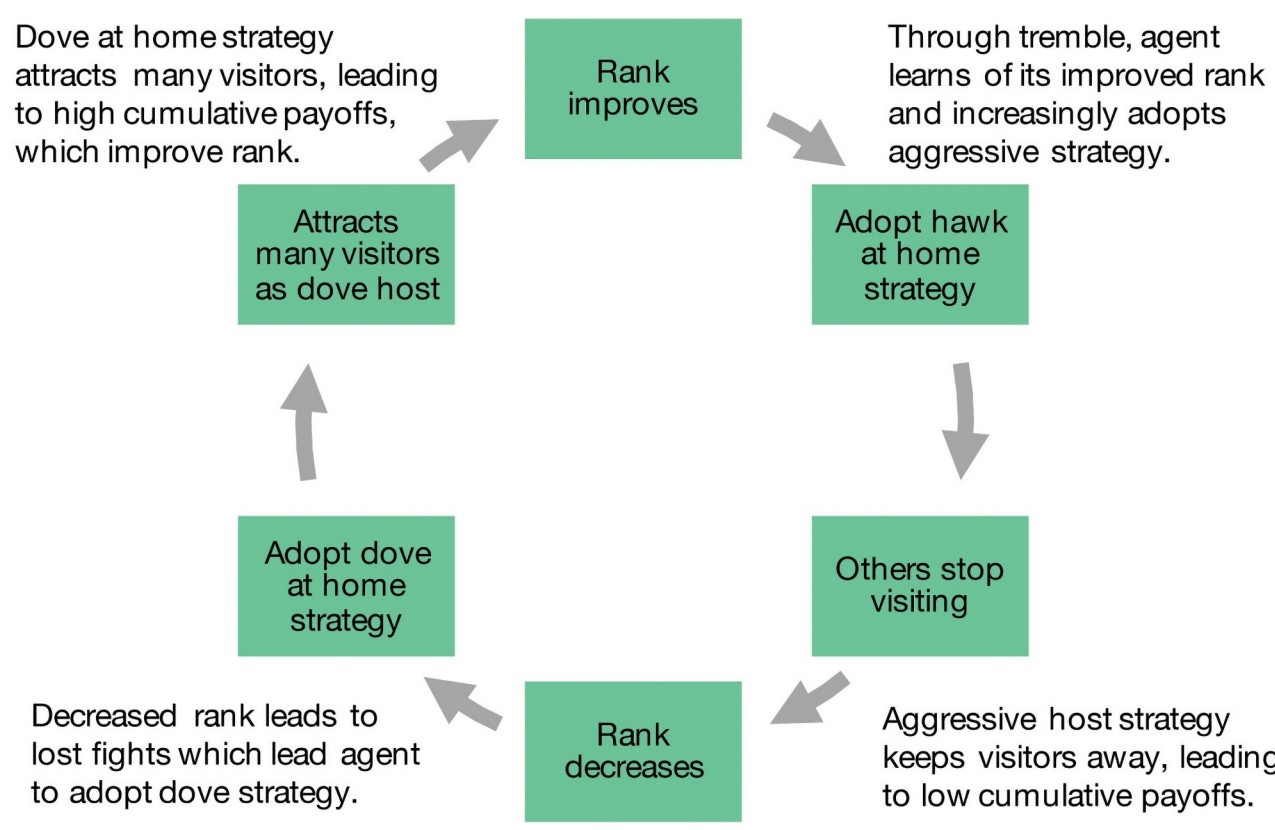

**Fig 4. Cycles of interacting rank and strategy changes.** Allowing individuals' ranks to change based on cumulative payoffs (dynamic ranks) leads to coupled cycles of aggressive-cooperative strategy, shifting network position, and rising and falling rank.

Do all individuals cycle through ranks or just some? We find that all individuals cycle through ranks and spend a similar amount of time at each rank (Fig C in S1 Text). Only the host strategy changes as individuals' ranks change, while the visiting strategy remains unaffected and is close to 100% *hawk*. This illustrates the mechanism behind the temporal dynamics: the interaction between partner choice and dynamic power asymmetry is crucial. Without partner choice, bullies cannot be isolated, and without dynamic power asymmetry, bullies do not lose their power even when isolated. So we only see complex temporal dynamics emerge with the two mechanisms together—partner choice and dynamic power. Since the payoffs are non-negative, individuals never strictly prefer to opt out of any interaction. This contributes to the rank cycling. Individuals are able to accumulate positive payoffs as hubs playing *dove* hosting many *hawk* visitors, thereby enabling their rise in the rankings. In effect, having more interactions tends to increase an individual's social ranking.

The length of the cycles depends on the degree of power asymmetry (Fig 5B). Below the critical value of $f < dh$, no cycles emerge as the game remains a mean game of conflict for all individuals. At the critical value $f \geq dh$, a phase transition occurs and cycles first emerge. Consequently, the same phase transition occurs in all three of our models (random mixing, partner choice, and dynamic ranks) at precisely $f \geq dh$. The higher the asymmetry in competitive ability, the shorter cycles become. When the benefits of winning fights are higher, ranking individuals adopt aggressive strategies more quickly. This in turn leads to isolation and a fall in the

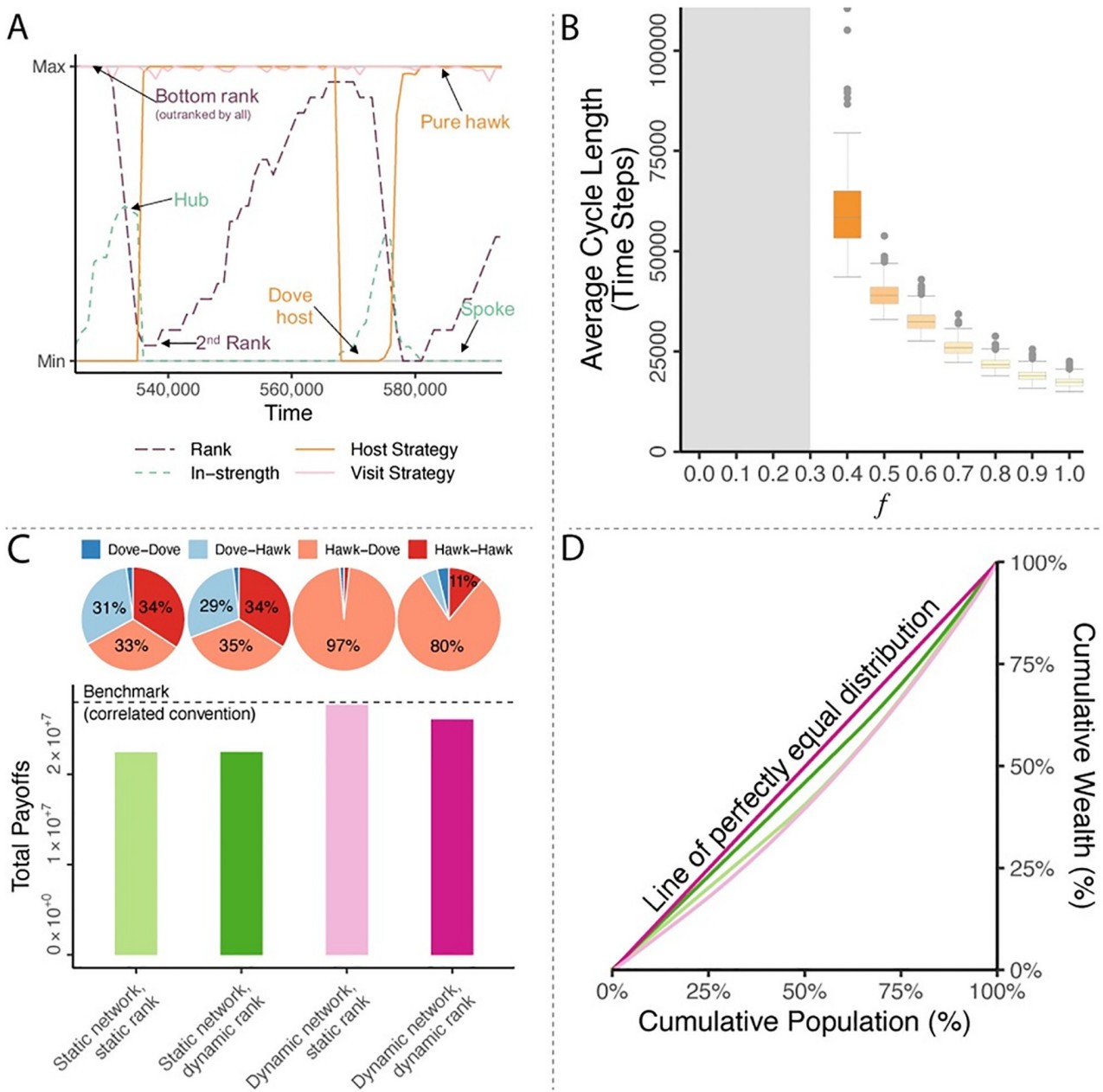

**Fig 5. Cycles have equalizing force that reduce wealth inequality. A**, Two example cycles of coupled changes in network positions, strategies, and ranks ($dh = 0.4$; $dd = 0.6$; $f = 0.4$). **B**, Below the critical value $f < dh$, ranks remain static and no cycles emerge (the grey shaded area). At $f \geq dh$, a phase transition occurs and coupled cycles emerge. The cycle length varies as a function of $f$. Cycles are shorter when power asymmetry is higher. **C**, Static networks lead to low total payoff due to inefficient *hawk-hawk* conflict while partner choice stabilizes the convention and leads to high payoffs. **D**, Static networks lead to high inequality as bullies earn higher payoffs. Partner choice with static ranks reverses the pattern with low-ranking individuals earning high payoffs as they attract more visitors. Partner choice and dynamic ranks preserving perfect equality among individuals. ($dh = 0.4$; $dd = 0.6$; $f = 0.6$).

rankings to happen more quickly. The result is shorter cycles. Beyond the effect of the degree of power asymmetry, timescales for partner choice updating also affect cycle length (Fig F in S1 Text). Cycles continue to emerge even with very slow partner updating, but they may be long and can even exceed our simulation timescales.

Our model also suggests several welfare and wealth inequality insights. Total population payoffs are lowest in random interaction case as several high-ranking individuals become bullies and break from the correlated convention (Fig 5C). This introduces significant friction through *hawk-hawk* conflict (31% of interactions), which reduces population welfare. The correlated convention is mostly restored with partner choice, leading to high population welfare. Individuals avoid visiting *hawk*-playing others who outrank them. Once the network settles, those collisions can be entirely avoided and may only be due to trembling. As a result, only a few *hawk-hawk* conflicts occur so that population welfare is close to the correlated convention equilibrium. This is different in the case of dynamic ranks. As individuals do not know each others' ranks before they visit, the welfare loss due to *hawk-hawk* conflict is larger. As some individuals adopt the *hawk*-at-home strategy as a result of their improved rank, others must then learn not to visit them, leading to some *hawk-hawk* conflict in the meantime (around 11% of total interactions). This drives down population welfare somewhat, but it remains higher than it is in the case without partner choice.

Wealth inequality is highest for static networks in which high-ranking individuals reap more rewards and further entrench their ranks, leading to a "the rich get richer" phenomenon (Fig 5D). Partner choice reduces inequality, but does not eliminate it completely. In fact, partner choice reverses the relationship between power and accumulated payoffs (Fig I in S1 Text). Now lower-ranked individuals reap more rewards as they attract more visitors: they are efficient cooperators who earn high rewards via frequent social interactions. When ranks are dynamic, the population achieves perfect equality over time as no individual remains a bully for long. Each individual cycles through ranks to even out payoffs.

In summary, even with these dynamic cycles occurring, the host-guest behavior of avoiding conflicts emerges and persists in the population. The broad pattern of behavior (the host-guest norm) is stable despite the fact that no individual remains static in rank, behavior, or social ties. As a result, costly *hawk-hawk* conflicts are rare, occurring only during transitions. Individuals learn to avoid others who become aggressive or to alter their behavior in response.

## Discussion

Understanding the emergence and stability of cooperative rules is critical to avoid the many paths that erode cooperation, including the tragedy of the commons [47–51]. Our model provides further insight into the evolution of cooperation among unequals. Cooperation in this context is understood as adopting rules or conventions to avoid conflict and adhering to them. In the hawk-dove game of conflict, correlating one's strategy on host and guest roles acts as a sort of egalitarian norm. Often such egalitarian behavior collapses in the context of hierarchies where dominant alphas can take what they want. Our study shows that partner choice can maintain the egalitarian behavior in the presence of power asymmetries. Such conventions would otherwise break down, as our random interaction model demonstrates. When an individual's hierarchical position depends on past competitive success and is modeled by treating rank as a function of cumulative payoffs, the egalitarian behavior tends to persist and we see dynamic cycles emerge. Despite the instability of the power dynamics and the cycles of bullies, the host-guest rule remains stable and resilient in most of the population. The presence of bullies ensures some individuals will adopt aggressive strategies, but the host-guest norm minimizes their impact since all individuals, including those of low ranks, are aggressive when visiting. The norm of deference to visitors prevents costly conflict throughout the population and, in conjunction with partner choice, cannot be destabilized by bullies. Interestingly, partner choice in the presence of power asymmetries allows only for host-guest rules and not ownership rules. Thus, whether the host-guest constellation in this case is properly called a

"convention" depends on the precise definition of convention, and whether conventions require viable alternative behavioral rules [13, 52].

Developing our work along three models (starting from the random interaction base case) is useful for comparative analysis. An important common feature of the three models is that the $f \geq dh$ transition point is robust. It marks the point at which the first ranking individual breaks from the convention in all three models. In the dynamic power asymmetry case it also marks the point at which cycles start to emerge: $f \geq dh$ is the only condition that needs to be satisfied. The differences in results between the models reveals the significance of dynamic networks, power asymmetries, and changing payoffs in understanding social interactions. Traditional analyses often assume fixed interaction structures and symmetric games with fixed payoffs. Our results show that the interdependence of these elements can have a major impact on dynamics. Thus, it is crucial to consider these interdependencies when attempting to generalize lessons from game theoretic and network models to real systems and to provide directions for future research.

An innovative feature of our modeling framework involves representing how a game may change over time [51, 53]. We analyze how the ranks that determine payoff asymmetry evolve as a function of cumulative payoffs. When individuals interact in social dilemma games over multiple rounds, it is typically assumed that any asymmetry that may exist between them is both exogenously determined and independent of the outcome of previous interactions [54–56]. Here we have introduced the idea that the source of asymmetry between individuals may dynamically depend on past payoffs. This reflects a change in the source of the ranking hierarchy from prior attributes to a more fluid source determined by the accumulation of capital. The contrast between static and dynamic ranking parallels the two main empirical hypotheses about the emergence of dominance hierarchies [57–60]. Payoffs in one round of our model affect payoffs in the next, as is the case in many real-world situations. Individuals in our model eventually learn (through trembling hands) that they outrank most of the population and resort to bullying, which leads to the cycles.

Our model uses relatively simple reinforcement learning mechanism. This implies that managing cooperative rules in the presence of bullies may be relatively easy to achieve as long as individuals are able to adjust their network ties to others. It also presumes that bullies may not create general obstacles that may change the strategic interactions, game payoffs, etc. When individuals cannot avail themselves of dynamic network connections—when they do not have the freedom to choose or change interaction partners—then bullies become more problematic. Most work on the destabilizing effects of bullies presumes this sort of environment [3–5]. Conversely, our results suggest that when power asymmetries are minor and/or network learning is slow relative to strategy learning (i.e., when individuals cannot very easily change their social networks), the cycles that emerge under dynamic power asymmetry are very long, to the point of effectively being absent. This could explain why we see "rich-get-richer" phenomena in the real world despite social networks (in principle) being dynamic.

Finally, the positive effect of partner choice has been recognized in evolutionary contexts, with a focus on games like the Prisoner's Dilemma [35, 61]. Our results reveal that this lesson generalizes to situations of conflict and convention. Thus, our model contributes to the growing body of work that explains network formation beyond preferential attachment and fitness models [22, 41]. Learning to avoid conflict with aggressive bullies in dynamic networks yields network structures that are at times qualitatively similar to those we observe in human social networks. The majority of network formation models do not involve strategic choice, but are a stochastic process of forming ties [22, 23], active linking [62] or assortative interactions [63]. Models that do allow for strategic choice assume that the basis on which choices are founded remains static. This results in stable (possibly unique) equilibrium networks [64–67]. None of

these models captures realistic dynamics such as more powerful individuals adopting more aggressive strategies and others may choosing to avoid interacting with them. None of the existing models can therefore explain the cycles we observe. Our work is part of a family of models being developed now that can explain richer patterns of network formation [68]. In particular, the realization that the maintenance of network ties requires effort and is dependent on the positive utility derived from those interactions.

## Methods

We use agent-based modeling which we analyze through computer simulation. We adapt the modeling framework developed by [12] but extend it with power asymmetries that are either static or dynamic. Except for diagrams that show examples, all results are averaged across 200 random seeds of the simulation. We model $N$ agents engaged in pairwise games of conflict with payoffs determined by the payoff matrix in Eq 1. All agents are ranked 1, 2, ..., $N$ according to their "fighting ability" with the ranking agent winning $f$ in aggressive *hawk-hawk* interactions while the outranked agent receives 0. That is, we implement a rank-order contest in which the ranking agent receives the entire $f$ payoff, irrespective of how large the difference in ranks is.

Agents interact through a dynamic network and simultaneously learn which strategy to play and whom to interact with by updating both strategy and network weights via Roth-Erev reinforcement [38]. The sequence of actions taken every round is as follows. First, each agent chooses one other agent to visit with a probability proportional to its outgoing network weights. Second, the visitor and the host agents chooses an action of either cooperate (*dove*) or defect (*hawk*) according to their visitor and host strategy weights, respectively. Third, each agent receives a payoff according to the payoff matrix. Fourth, all outgoing network weights and strategy weights are updated simultaneously for all agents via Roth-Erev reinforcement. Note that all agents are guaranteed exactly one interaction as visitor and between 0 and $N-1$ interactions as a host. Agents then interact repeatedly over 1 million rounds.

This model has two important types of asymmetries. First, agents can play different strategies depending on their role as host or visitor. Second, since network ties are directed, who an agent decides to visit can be different from who it is visited by. Together, these asymmetries allow for the possibility of correlated conventions to emerge. During the update step, hosts only update their strategy choices when hosting, whereas the visitors update both their network connection and strategy choices when visiting. The rationale for this particular asymmetry is that while individuals can control their behavioural strategies when hosting or visiting, and who they visit, they cannot control who decides to visit them.

**Strategy and partner choice updating**. Each agent $i$ has two vectors $(w_H, w_D)$ and $(w_h, w_d)$ giving $i$'s reinforcement weights for playing the *hawk* and *dove* strategy when hosting and visiting, respectively. We use two modular dynamical components commonly used in reinforcement learning [38] in our model: discounting of learned weights ($\delta$) and random trembling (errors, $\epsilon$). Trembling works identically for all actions (choosing a partner, choosing an action as visitor, choosing a response as host). The probability of choosing an action $s$ is proportional to the current relevant weights:

$$Pr(s) = (1 - \epsilon)\frac{w_s}{\sum_{s'} w_{s'}} + \epsilon\frac{1}{|S|} \tag{3}$$

where $\epsilon$ is the error rate, $S$ is the relevant set of available choices (i.e., visiting strategies, hosting strategies, or choice of player to visit), $s \in S$ and $s' \in S$.

All strategy weights are updated simultaneously for all agents at the end of each round with the received payoffs ($\pi$) and discounted by a factor $\delta$ (set at 0.01 unless otherwise noted) according to

$$w'_s = (1 - \delta)w_s + \pi_s \qquad (4)$$

where $w'_s$ is the weights after updating and $s$ is the action that was chosen; $\pi_s = 0$ if action $s$ was not chosen. All visits and strategy choices occur simultaneously within a round and all weights are updated simultaneously after each round. Agents keep separate strategy weights for hosting and visiting and this asymmetry is maintained during the updating process such that the payoffs received for hosting do not affect strategy weights for visiting or vice versa.

Each agent $i$ has a vector representing their directed weighted network ties with other agents $(w_{i1}, w_{i2}, \ldots, w_{in})$ where $w_{ij}$ represents the weight that agent $i$ chooses to visit agent $j$; $(w_{ii} = 0)$. We initialize weights as 1 for both strategy choices and network weights uniformly as $\frac{L}{N-1}$. To standardize learning speeds between strategy updating and network updating, we set $L = 19$ which yields starting weights of 1 for all network ties in our standard population of $N = 20$ ($w_{ij} = 1$ for all $i, j$ and $i \neq j$). Keeping $L$ constant across populations of different sizes allows us to keep the reinforcement learning at a similar speed relative to total initial weights.

When selecting an interaction partner for a given round, the probability of choosing agent $j$ is proportional to agent $i$'s outgoing network weights as

$$Pr(j) = (1 - \epsilon)\frac{w_{ij}}{\sum_k w_{ik}} + \epsilon \frac{1}{|N|} \qquad (5)$$

where $\epsilon$ is the error rate, $N$ is the set of agents, $j \in N$ and $k \in N$.

After each round of interactions, the weights for all outgoing partner choice links are updated by discounting the prior weight by a factor ($\delta$) and adding the received payoff ($\pi$)

$$w'_{ij} = (1 - \delta)w_{ij} + \nu\pi_{ij} \qquad (6)$$

where $w'_{ij}$ is the link weight after updating and $\pi_{ij}$ is the most recent payoff and $\nu$ is an adjustment factor for the timescale of network learning. We investigate differences in relative learning speeds by changing the network learning speed relative to the speed of strategy learning. We modify the speed of network learning by multiplying the payoffs received after each round by an adjustment factor $\nu$ before updating network weights but leaving payoffs unmodified for strategy updating. We investigate network updating both at speeds slower than strategy updating (e.g., $\nu = 0.1$) and faster than the strategy updating speed (e.g., $\nu = 10$).

**Rank updating**. We consider two versions of this game: static and dynamic. In the static case, each agent is randomly assigned a rank between $1, \ldots, N$ at the beginning which remains the same throughout all interactions. Ranks are private information but players learn the ranks of others through reinforcement learning. The payoff for the ranking agent $i$ playing *hawk* against an outranked agent $j$ also playing *hawk* is $f$, which we vary from 0 (ranks effectively do not matter at all) to 1.0 (ranks matter greatly), while the outranked agent always receives a payoff of 0.

In the dynamic ranks case, we update ranks based on cumulative payoffs every 1, 000 rounds. Updating ranks every 1, 000 rounds avoids noise from tremble and updating ranks more/less often does not substantively alter our our results. In case of a *hawk-hawk* interaction among agents with identical cumulative payoffs (identical cumulative payoffs are extremely rare given discounting but can happen in early rounds) each agent receives a payoff of $f/3$, which represents an equal chance of winning the resource but with some cost of conflict.

Agents who are successful in their interactions (achieve high payoffs) and attract many visitors, accumulate more payoffs and are better ranked.

## Supporting information

**S1 Text. Supporting information.** Supporting information containing additional description of game notation, review of related work (Table A), and mathematical analyses of the static game, the dominant solvable breakpoint, and the breakdown of conventions. It also contains additional simulation results and additional figures for evolution of strategies (Fig A), network evolution (Fig B), partner choice and dynamic ranks (Fig C), comparison of timescales (Fig D and Fig E), cycle length (Fig F), total payoffs (Fig G), inequality (Fig H), payoffs by rank (Fig I), and cycle length for different population sizes (Fig J).
(PDF)

## Acknowledgments

We are grateful for constructive comments by Zach Fulker.

## Author Contributions

**Conceptualization:** Rory Smead, Patrick Forber, Christoph Riedl.

**Data curation:** Michael Foley.

**Formal analysis:** Rory Smead, Patrick Forber.

**Funding acquisition:** Christoph Riedl.

**Investigation:** Michael Foley, Rory Smead, Patrick Forber, Christoph Riedl.

**Methodology:** Rory Smead, Patrick Forber, Christoph Riedl.

**Project administration:** Christoph Riedl.

**Resources:** Christoph Riedl.

**Software:** Michael Foley, Christoph Riedl.

**Supervision:** Christoph Riedl.

**Validation:** Michael Foley, Rory Smead, Patrick Forber, Christoph Riedl.

**Visualization:** Michael Foley, Christoph Riedl.

**Writing – original draft:** Rory Smead, Patrick Forber, Christoph Riedl.

**Writing – review & editing:** Michael Foley, Rory Smead, Patrick Forber, Christoph Riedl.

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
