## [Decision Letter · Decision Letter 0]

2 Dec 2020

Dear Prof. Dr. Riedl,

Thank you very much for submitting your manuscript "Avoiding the bullies: Resilience of cooperation among unequals" for consideration at PLOS Computational Biology.

As with all papers reviewed by the journal, your manuscript was reviewed by members of the editorial board and by several independent reviewers. In light of the reviews (below this email), we would like to invite the resubmission of a significantly-revised version that takes into account the reviewers' comments.

Thanks very much for sending your work to PLoS CB. All reviewers like the manuscript and think it a good fit for the journal. Although they cannot recommend it published in present form, they would like to see a revised version where their extensive comments are satisfactorily addressed. We are looking forward to receiving your revised manuscript.

We cannot make any decision about publication until we have seen the revised manuscript and your response to the reviewers' comments. Your revised manuscript is also likely to be sent to reviewers for further evaluation.

Sincerely,

Feng Fu

Guest Editor

PLOS Computational Biology

Natalia Komarova

Deputy Editor

PLOS Computational Biology

Thanks very much for sending your work to PLoS CB. All reviewers like the manuscript and think it a good fit for the journal. Although they cannot recommend it published in present form, they would like to see a revised version where their extensive comments are satisfactorily addressed. We are looking forward to receiving your revised manuscript.

Reviewer's Responses to Questions

**Comments to the Authors:**

Reviewer #1: Please see attached review.

Reviewer #2: In this paper, the authors aim to study the evolution of cooperation behavior (egalitarian norms/conventions) in the presence of dominant bullies (power asymmetry) using a theoretical framework of evolutionary game theory, in this case, a generalized hawk-dove game of conflict. They find the cooperative conventions can always be maintained if the system allows partner choice, letting individuals change their interaction partners via a simple reinforcement learning process, whereas such conventions would break down if individuals cannot choose their partner freely (such as the random interaction case). Further, they introduce dynamic power asymmetries where individual’s rank depends on the cumulative payoffs and find the emergence of evolutionary cycles of coupled network-strategy-rank changes. The main innovation lies on the incorporation of both power asymmetry and partner choice into games of conflict, and the authors have explicitly demonstrated the nonnegligible power of these two mechanisms in persisting cooperation among unequals.

Overall, I have enjoyed reading this clearly written manuscript, with well-established motivations in an informative introduction and solid results provided by both the main text and the supplementary information. The mathematical derivations are sound and confirmed by sufficient agent-based simulations. The topic is important and attractive. And the model shows a rich set of evolutionary behaviors, some of which are counterintuitive and indeed interesting. In particular, I like the in-depth Discussion that explicitly demonstrates the important insights from the model. I believe this work provides a valuable contribution for the study of cooperation behaviors and will arouse great attention in many fields. Thereby, I am happy to recommend the acceptance of this paper in its current form.

Meanwhile, I would like to provide some minor suggestions that the authors may consider for a further improvement:

(1) The first two sentences in Introduction are exactly the same as in Abstract. It’s better to use a different expression.

(2) The resolution of Fig.1 b is poor in the current version. Also, it’s better to add what the node size represents in the caption of Fig.1 b.

(3) I find the non-monotonous phenomenon in Fig 3 a and b where the highest proportion of pure hawks emerges at intermediate values of f is interesting and I notice that the authors have tried to provide an intuitive explanation (Line 220-221, 225-227). However, I feel that it can be better explained as a combined effect of both avoiding conflicts with ranking individuals and avoiding less benefits from hawk-dove interactions with lower-ranked individuals. The latter is caused by partner choice: lower-ranked individuals would reduce visits to a higher-ranked Hawk host.

(4) some minor mistakes:

Line 18: is dangerous work -- is a dangerous work

Line 76: in that that -- in which

Line 226: at least at least -- at least

Line 615: when does does – when does

Reviewer #3: Foley et. al. investigated whether or not egalitarian conventions survive the presence of dominant individuals that are ensured victory in conflicts. To this end, they focus on the evolutionary dynamics of Hawk-dove game with unequals on dynamical networks. They show that the interaction between social power asymmetry (visit or host) and the partner choice arising from the dynamical nature of the social network, plays a crucial role in giving rise to such egalitarian norms. In addition, they show two other models cannot give rise to such resilience egalitarian convention (with asymmetry and without evolving network, without symmetry and with evolving network). Therefore, it suggests that the interaction between asymmetry and partner choice is key to egalitarian conventions.

The results are sound and this work can have a wide audience in PLOS C. B., since cooperation is an important topic in evolutionary biology. However, the paper is not well-organized and technically I also have some concerns. Therefore, I recommend a major revision.

Comments:

1) Time scales. There are several dynamics: the rank and the network, and the strategy. Previous studies show [66,69] that the time scale of the strategy dynamics and the underlying network would greatly alter the cooperation level. In this work in the method, the authors only adopt one time scale (line 513) for rank dynamics. How Robust is the result with the time scale of different dynamics? A discussion along this line would benefit the audience.

2) Fig 1 is not illustrative enough, I am not clear from the figure how network evolves, and how individuals adjust their behaviors.

3) “Convention” is not defined in a clear way. Typically convention refers to the believing system (e.g. A thinks that B is good if B helps C), here, it seems to refer to a resulting population composition. It should be clearly defined.

4) The interaction between asymmetry and network paves the way (intuitively) to an oscillatory dynamics. (line 385-391), how on earth that this oscillatory dynamics lead to an resilient convention? An intuitive understanding would benefit.

**Have all data underlying the figures and results presented in the manuscript been provided?**

Reviewer #1: Yes

Reviewer #2: None

Reviewer #3: Yes

PLOS authors have the option to publish the peer review history of their article (what does this mean?). If published, this will include your full peer review and any attached files.

Reviewer #1: No

Reviewer #3: No
---

## [Decision Letter · Decision Letter 1]

2 Mar 2021

Dear Prof. Dr. Riedl,

We are pleased to inform you that your manuscript 'Avoiding the bullies: The resilience of cooperation among unequals' has been provisionally accepted for publication in PLOS Computational Biology.

Best regards,

Feng Fu

Guest Editor

PLOS Computational Biology

Natalia Komarova

Deputy Editor

PLOS Computational Biology

I would like to concur with the unanimous recommendation of the three reviewers to accept your excellent contribution to PLoS Computational Biology. As you will see, Reviewer 1 has made some optional comments, which I hope you would be able to take into account when finalizing the accepted version of the manuscript. Also it would be helpful to provide the repository DOI at Harvard Dataverse as said in the cover letter for making simulation codes pulbicly available. Last but not least, the supplementary information is currently appended right after the main text (where SI figures should have been labelled Fig.Sx properly). Thank you for your consideration and congratulations on the nice work!

Reviewer's Responses to Questions

**Comments to the Authors:**

Reviewer #1: I would like to thank the authors for all of their effort incorporating suggestions from first round of review. I believe that the manuscript has been much improved, and, in particularly, Figure 1 and the explanation of the model is much clearer in the revised submission. I wholeheartedly support the acceptance of this paper in PLoS Computational Biology. However, i do have a few minor comments that I believe could help for the published version of the paper.

1. I believe that the new simulations in which individuals playing dove against those playing hawk cannot accumulate payoff are helpful for understanding the mechanisms driving the cycling behavior when rankings can change according to payoff. If the authors feel that the mechanism for mimicking negative payoff is too artificial, it may be possible to see similar behavior with non-negative payoffs if the hawk-dove payoff is greater than n-1 times the dove-hawk payoff. In that case, even an individual playing dove against a whole population of hawk-playing visitors would not be able to rise in the rankings. Either way, I think that the behavior in the case in which playing a lot of games and "losing" all of them is worse than playing few games but "winning" is worth mentioning in some form in the supplement.

2. I think it could be useful for the authors to mention that the question of whether to consider static versus dynamical rankings is quite relevant to the literature on the formation of dominance hierarchies. In particular, both "prior attributes" (some underlying strength or skill) or "social winner/loser effects" (dominance rankings depending on past performance) have been posited to explain the structure and formation of social hierarchies. I have included below several references related to mathematical and experimental approaches to this question.

3. In the sentence "Since the payoffs are ≥ 0", I think the word "non-negative" should be used instead of the symbol "≥".

References

Chase, I. D., Tovey, C., Spangler-Martin, D., & Manfredonia, M. (2002). Individual differences versus social dynamics in the formation of animal dominance hierarchies. Proceedings of the National Academy of Sciences, 99(8), 5744-5749.

Dugatkin, L. A. (1997). Winner and loser effects and the structure of dominance hierarchies. Behavioral Ecology, 8(6), 583-587.

Franz, M., McLean, E., Tung, J., Altmann, J., & Alberts, S. C. (2015). Self-organizing dominance hierarchies in a wild primate population. Proceedings of the Royal Society B: Biological Sciences, 282(1814), 20151512.

Laskowski, K. L., Wolf, M., & Bierbach, D. (2016). The making of winners (and losers): how early dominance interactions determine adult social structure in a clonal fish. Proceedings of the Royal Society B: Biological Sciences, 283(1830), 20160183.

Reviewer #2: The authors have addressed all my previous comments, and I am satisfied with their responses. In particular, the logical framework of the entire manuscript looks much better due to the revisions on introduction, Fig. 1 and structural adjustments. And the authors have also made significant efforts to remove repetition and fix usage and grammar, which improves the overall readability.

Therefore I recommend accepting it for publication in PLOS Computational Biology.

Reviewer #3: The authors have replied to all of my concerns, and the manuscript is much clearer than the last submitted version. Thus I recommend the publication of the manuscript in the current form.

**Have all data underlying the figures and results presented in the manuscript been provided?**

Reviewer #1: Yes

Reviewer #2: Yes

Reviewer #3: Yes

PLOS authors have the option to publish the peer review history of their article (what does this mean?). If published, this will include your full peer review and any attached files.

Reviewer #1: No

Reviewer #2: **Yes: **Xin Wang

Reviewer #3: No

---

## [Editor Report · Acceptance letter]

4 Apr 2021

PCOMPBIOL-D-20-01822R1 

Avoiding the bullies: The resilience of cooperation among unequals

Dear Dr Riedl,

I am pleased to inform you that your manuscript has been formally accepted for publication in PLOS Computational Biology. Your manuscript is now with our production department and you will be notified of the publication date in due course.

With kind regards,

Alice Ellingham
